# Role of Advanced Gastrointestinal Endoscopy in the Comprehensive Management of Neuroendocrine Neoplasms

**DOI:** 10.3390/cancers15164175

**Published:** 2023-08-19

**Authors:** Harishankar Gopakumar, Vinay Jahagirdar, Jagadish Koyi, Dushyant Singh Dahiya, Hemant Goyal, Neil R. Sharma, Abhilash Perisetti

**Affiliations:** 1Department of Gastroenterology and Hepatology, University of Illinois College of Medicine at Peoria, Peoria, IL 61605, USA; hgopakum@uic.edu; 2Department of Internal Medicine, University of Missouri-Kansas City, Kansas City, MO 64110, USA; jahagirdarv@umkc.edu (V.J.); jagadishkoyi@umkc.edu (J.K.); 3Division of Gastroenterology, Hepatology & Motility, The University of Kansas School of Medicine, Kansas City, KS 66160, USA; dush.dahiya@gmail.com; 4Department of Surgery, Center for Interventional Gastroenterology at UT (iGUT), The University of Texas Health Science Center, Houston, TX 77054, USA; doc.hemant@yahoo.com; 5Advanced Interventional Endoscopy & Endoscopic Oncology (IOSE) Division, GI Oncology Tumor Site Team, Parkview Cancer Institute, 11104 Parkview Circle, Suite 310, Fort Wayne, IN 46845, USA; nrsharma219@yahoo.com; 6Division of Gastroenterology and Hepatology, Kansas City Veteran Affairs, Kansas City, MO 64128, USA

**Keywords:** neuroendocrine tumors, endoscopic ultrasound, NETs, PNETs, neuroendocrine neoplasms, gastropancreatic neuroendocrine tumors, endoscopy, endoscopic resection, endoscopic ablation

## Abstract

**Simple Summary:**

Neuroendocrine tumors (NET) arise from cells throughout the diffuse endocrine system. They are primarily sporadic but can also occur in the context of genetic syndromes like multiple endocrine neoplasia. They are most commonly seen in the gastrointestinal tract, pancreas, and lungs. Data show that the incidence of NETs is increasing, partly contributed to by improving diagnostic modalities and their increasing use. The management of larger (>2 cm) NETs and those with evidence of local or metastatic involvement is relatively well defined and involves surgery, chemotherapy, or a combination. Endoscopic evaluation is pivotal in diagnosing, staging, and grading NETs. Furthermore, advanced endoscopic techniques like resection and ablation can now successfully treat NETs, particularly those less than 2 cm, potentially reducing the adverse events and healthcare costs associated with surgical management. This article discusses the latest advances in the endoscopic evaluation and management of this rare condition.

**Abstract:**

Neuroendocrine neoplasms (NENs), also called neuroendocrine tumors (NETs), are relatively uncommon, heterogenous tumors primarily originating in the gastrointestinal tract. With the improvement in technology and increasing use of cross-sectional imaging and endoscopy, they are being discovered with increasing frequency. Although traditionally considered indolent tumors with good prognoses, some NENs exhibit aggressive behavior. Timely diagnosis, risk stratification, and management can often be a challenge. In general, small NENs without local invasion or lymphovascular involvement can often be managed using minimally invasive advanced endoscopic techniques, while larger lesions and those with evidence of lymphovascular invasion require surgery, systemic therapy, or a combination thereof. Ideal management requires a comprehensive and accurate understanding of the stage and grade of the tumor. With the recent advancements, a therapeutic advanced endoscopist can play a pivotal role in diagnosing, staging, and managing this rare condition. High-definition white light imaging and digital image enhancing technologies like narrow band imaging (NBI) in the newer endoscopes have improved the diagnostic accuracy of traditional endoscopy. The refinement of endoscopic ultrasound (EUS) over the past decade has revolutionized the role of endoscopy in diagnosing and managing various pathologies, including NENs. In addition to EUS-directed diagnostic biopsies, it also offers the ability to precisely assess the depth of invasion and lymphovascular involvement and thus stage NENs accurately. EUS-directed locoregional ablative therapies are increasingly recognized as highly effective, minimally invasive treatment modalities for NENs, particularly pancreatic NENs. Advanced endoscopic resection techniques like endoscopic submucosal dissection (ESD), endoscopic submucosal resection (EMR), and endoscopic full-thickness resection (EFTR) have been increasingly used over the past decade with excellent results in achieving curative resection of various early-stage gastrointestinal luminal lesions including NENs. In this article, we aim to delineate NENs of the different segments of the gastrointestinal (GI) tract (esophagus, gastric, pancreatic, and small and large intestine) and their management with emphasis on the endoscopic management of these tumors.

## 1. Introduction

Neuroendocrine neoplasms (NENs) are a heterogeneous group of malignancies arising from the “neuroendocrine” cells of the diffuse endocrine system and dispersed throughout the endocrine system and the endocrine islet tissue embedded in glandular tissue such as those in the gastrointestinal, pancreatic, and respiratory tracts [1,2]. A recent study reported a rising annual incidence of NENs in the United States from 1.09 per 100,000 persons in 1973 to 6.98 per 100,000 persons in 2012 based on an analysis of the Surveillance, Epidemiology, and End Results (SEER) database [3]. The rising incidence has been attributed at least partially to earlier detection of asymptomatic disease due to improvements, the availability, and the increased utilization of advanced diagnostic technologies [3]. The 2019 WHO classification (Table 1) broadly divided NENs into neuroendocrine tumors (NETs) and neuroendocrine carcinomas (NECs) based on their molecular differences [4]. Classification of the tumors is based on grade, mitotic rate, and Ki67 index, as numerous studies have confirmed that NENs with increased mitotic rate and a high Ki-67 index are associated with a more aggressive clinical course and worse prognosis [2]. Well-differentiated tumors were subclassified into low grade (NET G1, mitotic rate < 2 mitoses/2 mm^2^, Ki67 index < 3%), intermediate grade (NET G2, mitotic rate 2 to 20 mitoses/2 mm^2^, Ki67 index 3–20%), or high grade (NET G3, mitotic rate > 20 mitoses/2 mm^2^, Ki67 index > 20%) [4]. Poorly differentiated high-grade tumors (neuroendocrine carcinoma or NECs) were subclassified as small-cell type (SCNEC) or large-cell type (LCNEC), where both have > 20 mitotic rate and >20% Ki67 index. Mixed neuroendocrine-non-neuroendocrine neoplasms (MiNENs) were either well or poorly differentiated with variable grading, mitotic rate, and Ki67 index [4]. NENs are staged using the American Joint Committee of Cancer (AJCC), tumor (T), node (N), and metastasis (M) staging system with the latest revisions in its eighth edition [5].

Gastrointestinal endoscopy has undergone tremendous advancements over the past two decades, positioning it as a minimally invasive, highly effective, and relatively less expensive diagnostic and therapeutic modality for various luminal and extraluminal pathologies in the abdomen and thorax. Today’s advanced endoscopes with high-definition white light imaging and image enhancement techniques like narrow band imaging (NBI) have improved the sensitivity and specificity of endoscopic diagnosis of gastrointestinal and biliary luminal lesions. Endoscopic ultrasound (EUS), introduced as a diagnostic tool, has now evolved into a powerful instrument capable of performing various advanced diagnostic and therapeutic interventions in a minimally invasive manner for conditions that traditionally require surgery. With the refinements in EUS needles and tissue diagnostics, EUS-guided fine needle aspiration and biopsy (EUS-FNA/EUS-FNB) now offers the ability to obtain high-quality tissue samples for accurate diagnosis and staging of conditions of both gastrointestinal luminal and extraluminal pathologies. EUS-guided fine needle injection (EUS-FNI), EUS-guided radiofrequency ablation (EUS-RFA), EUS-guided photodynamic therapy (EUS-PD), and EUS-guided microwave ablation are various options that can be used for treatment for malignant and pre-malignant conditions. These modalities have already shifted the paradigm in managing conditions like Barrett’s esophagus. Advanced resection techniques like endoscopic mucosal resection (EMR), including various modifications of EMR like underwater EMR, cap-assisted EMR, ligation-assisted EMR, endoscopic mucosal dissection (ESD), full-thickness endoscopic resection (FTER), and submucosal tunneling endoscopic resection (STER), are all part of an advanced endoscopist’s armamentarium to definitively manage various potentially malignant lesions of the luminal gastrointestinal tract. As most NENs are found in gastroenteropancreatic locations, the role of an advanced endoscopy in managing these conditions continues to increase and evolve. This approach offers a minimally invasive, highly effective, and often curative treatment in well-selected patient populations. Table 2 summarizes the various endoscopic diagnostic, staging, and therapeutic modalities available for the comprehensive evaluation and management of NENs. Table 3 summarizes the guidelines on the endoscopic management of NENs from the North American Neuroendocrine Tumor Society (NANETS) and the European Neuroendocrine Tumor Society (ENETS).

## 2. Role of Endoscopy in the Diagnosis, Staging, and Management of Nets Based on Location

### 2.1. Esophagus

Esophageal NENs (e-NENs) are rare, accounting for 0.2% to 1.3% of all GI NENs [6,7]. Lee et al. identified only 26 esophageal NENs among 2037 GI NETs [7]. Studies report a mean patient age of 60 years, with male preponderance [7,8]. Dysphagia, weight loss, and abdominal discomfort are common presenting symptoms [7,9]. NENs are sometimes discovered incidentally on esophagogastroduodenoscopy (EGD). They can range from low-grade carcinoid tumors, which have a good prognosis following resection, to high-grade NENs and large-cell or small-cell esophageal carcinomas that present as fungating masses [9]. Carcinoid syndrome is rarely seen on presentation since most tumors have a low degree of differentiation [9].

During EGD, low-grade e-NENs are generally seen as a single lesion in the lower third esophagus [7]. Esophageal neuroendocrine cancers (NECs) may be seen as exophytic polypoid masses, with or without ulceration and surface necrosis [10]. On white light imaging, surface redness may be seen due to vascularity. Narrow band imaging (NBI) may show abundant reticular vessels in carcinoid tumors. There is no specific TNM staging system for esophageal NECs owing to their rarity compared to esophageal squamous cell carcinoma and adenocarcinoma. They may be staged as ‘limited disease’ (LD) when confined to the esophagus or ‘extensive disease’ (ED) when they have spread beyond locoregional boundaries per the Veteran’s Administration Lung Study Group (VALSG) criteria [11]. Some authors used the 2009 American Joint Committee on Cancer (AJCC) TNM staging for esophageal squamous cell carcinoma for esophageal NECs [8]. Around 58% of esophageal NECs have evidence of regional lymph node involvement or widespread metastasis at diagnosis, with the liver, lung, and bone being the usual sites of distant spread [12].

EUS plays an essential role in identifying regional metastasis and estimating the extent of esophageal wall invasion for staging. Due to their rich vascularity, NENs are hyper-enhancing on contrast-enhanced EUS (CE-EUS). This hyper-enhancing nature differentiates them from hypo-enhancing leiomyomas, the most common esophageal smooth muscle tumor. Park et al., in their prospective study, reported that EUS-FNB was significantly faster and needed fewer samples than unroofing biopsy in diagnosing upper GI subepithelial tumors (SETs) (6 of 39 patients had esophageal SETs) [13]. A recent study concluded that EUS-FNA and key-hole biopsy (KHB) were equally effective in diagnosing upper GI SETs; however, the tissue sample was inadequate for determining the mitotic index [14]. Sanaei et al. found that EUS-FNB and single-incision with needle knife (SINK) were equally effective for upper GI SET sampling [15]. Mucosal incision-assisted biopsy (MIAB) is an upcoming technique, with a recent meta-analysis concluding that MIAB can be equally safe and effective as EUS-guided tissue acquisition [16].

A few case reports describe endoscopic management of esophageal NENs via polypectomy and endoscopic resection [17,18]. Definitive treatment guidelines for NETs have not yet been defined. Lee et al. proposed endoscopic resection (ER) for well-differentiated elevated tumors < 1 cm without regional lymph node metastasis or lymphovascular invasion [7]. Yagi et al. also similarly noted that ER could be considered for esophageal NENs < 10 mm in diameter, without ulceration or erosion, that are above the submucosal layer, given their low chance of lymph node metastasis [18]. Esophageal submucosal tumors (SMTs) can be safely resected by conventional endoscopic submucosal dissection (ESD) techniques, keeping in mind that the dissecting layer is thinner than in the case of epithelial lesions [19]. In their case series of 24 cases of foregut NEN lesions (1 in the esophagus, 24 in the stomach, and 4 in the duodenum), Li et al. reported histologically complete resection in 97% of cases using ESD [20]. The high R0 resection rate can help in accurate histological grading and staging and lower the chance of recurrence. In this series, there was no report of perforation, and only 1 case of delayed bleeding showing ESD can be a safe option for small e-NENs. Given the risk of perforation when ESD is performed for a submucosal tumor arising from the muscularis propria, Shim and Jung suggested that the depth of the lesion be evaluated first by EUS before treatment [21]. Endoscopic submucosal resection (ESD) may be preferred over endoscopic mucosal resection (EMR) as it allows for complete tumor resection with adequate horizontal and vertical margins. Due to the high en bloc resection rates, ESD also preserves the sample orientation for pathological assessment and has a lesser postoperative stricture risk [18]. Endoscopic enucleation with submucosal tunning (SMT) retains the mucosa and muscularis mucosa, reducing the chance of stricture formation [22].

Although endoscopic management can be the primary modality for the management of small esophageal NENs, larger tumors (>1 cm) warrant consideration for surgical resection and or chemotherapy. This is based on findings of increased rates of lymph node metastasis in NENs larger than 1 cm. However, given the rarity of esophageal NENs, these recommendations are based on the extrapolation of data from NENs in other locations of the GI tract. Esophageal NENs have high rates of being neuroendocrine carcinomas, and ESGE recommends treating them as esophageal adenocarcinomas [23]. Currently, there are no treatment guidelines that specifically address esophageal NENs, and treatment choice should be based on a comprehensive assessment of tumor size, grade, stage, the patient’s coexisting health conditions, and local expertise [24]. Future studies can be directed at analyzing the efficacy of EUS in identifying margins of NET before resection to ensure complete resection with clear boundaries. Patterns on CE-EUS for esophageal NEC can also be examined. Appropriate selection criteria for endoscopic resection are the need of the hour.

### 2.2. Stomach

Gastric neuroendocrine neoplasms (g-NENs) are subdivided into three types based on etiology, pathogenesis, and prognosis. Type 1 and Type 2 gastric NENs result from neuroendocrine cell hyperplasia from pathologically elevated gastrin levels [2]. Type 1 g-NENs, which represent 70% to 80% of all g-NENs, occur in the setting of chronic atrophic gastritis (autoimmune gastritis and *Helicobacter pylori*-associated gastritis), resulting in chronic achlorhydria and resulting hypergastrinemia. Type 2 g-NEN is from hypergastrinemia due to pancreatic or duodenal gastrinoma (Zollinger–Ellison syndrome) and can be seen in patients with multiple endocrine neoplasia type-1 (MEN1) [2]. Persistent hypergastrinemia exerts a proliferative effect on the enterochromaffin-like (ECL) cells in the stomach. Type 1 and Type 2 gastric NENs usually follow an indolent course with a low risk of progression or metastasis. Most type 1 and type 2 g-NENs are histological grade, G1-G2, with a risk of metastatic disease of about 2–5% for type 1 and 10–30% for type 2, depending on various factors [25]. The risk for metastasis increases significantly when the size exceeds 1 cm [23]. Most Type 3 g-NENs, on the other hand, are sporadic, occur in the absence of hypergastrinemia; are histological grade G1, 2, or 3; NECs or mixed type; and are aggressive with local or hepatic metastasis present in up to 65% of patients on presentation.

Gastric NENs are primarily diagnosed incidentally following endoscopy for non-specific symptoms such as dyspepsia, evaluation of anemia, or occult gastrointestinal bleeding. Type-1 g-NENs are typically small (<10 mm), often multiple, and are seen primarily in the gastric fundus or corpus [26]. Endoscopically, they appear as smooth, rounded, or polypoid submucosal lesions that can be yellowish or red in appearance (Figure 1) [23,26]. High-resolution magnifying endoscopy and narrow-band imaging (NBI) offer further characterization of these tumors. Most of the g-NEN surface is covered by normal mucosa, but often, a central depression can be seen where the gastric glands vanish. In this region, abnormally dilated subepithelial vessels with blackish-brown capillaries can be visualized under NBI imaging [26]. Type 2 g-NENs are similar to Type 1 g-NENs, but the adjacent gastric mucosa exhibits hypertrophic changes and often has coexisting areas of gastric or duodenal ulcerations. Type 3 gastric NENs usually occur as single, large (>2 cm) lesions that are often ulcerated.

EUS plays a pivotal role in the diagnosis and staging of g-NENs. The European Neuroendocrine Tumor Society (ENETS) recommends EUS evaluation for all g-NENs 1 cm or larger [27]. On EUS, Type 1 and Type 2 g-NENs are commonly seen in the second (deeper mucosal) or third (submucosal) echo layer and have a hypoechoic intramural structure [26]. They usually have a well-defined smooth surface and are round or oval with a uniform echotexture. In Type 2 g-NENs, EUS can also be used to evaluate for the presence of any pancreatic or duodenal gastrinomas.

Endoscopic resection (ER) is a safe and effective treatment option for low-grade g-NENs < 20 mm without evidence of locoregional or distant metastasis. Lesions > 20 mm in size have a higher risk for invasion and metastasis; hence, surgery is currently the preferred approach. The National Comprehensive Cancer Network (NCCN) guidelines recommend treating type I g-NENs 20 mm or less by local resection (endoscopic or surgical wedge resection) whenever feasible [2,28]. Both EMR and ESD have been used for this indication with excellent success. Appropriate patient selection is the key to favorable long-term outcomes following endoscopic resection for g-NENs. As discussed above, EUS is vital in evaluating the depth of invasion and locoregional spread to determine candidacy for endoscopic resection. Most of the currently available data are from studies evaluating the role of ER for type 1 g-NENs. A multicenter study on 187 patients from Taiwan showed an R0 resection rate of 100% with ESD and EMR for G1 and G2 g-NENs up to 20 mm, although the number of g-NENs > 10 mm was only three in this study [29]. A study that compared EMR to ESD for type 1 g-NENs found that although horizontal margins were negative regardless of technique, 66.7% of patients who underwent EMR had positive vertical margins [30].

In a study that evaluated the long-term outcomes of foregut NENs undergoing ER (EMR and ESD) or surgical resection, the non-curative resection rate was 24.2% in the ER group and 25% in the SR group. This study found no evidence of recurrence, even in cases with positive margins, and the only cases that progressed to metastatic disease were NECs with evidence of lymphovascular invasion [31]. This suggests that size should not be an absolute deterrent to ER, and a careful evaluation for lymphovascular invasion should determine the ideal resection technique and further management. As technology and experience with ER continue to grow, newer ER techniques like endoscopic full-thickness endoscopic resection (EFTR) using a full-thickness resection device (FTRD) and underwater EMR have been evaluated with promising results [32,33].

Surgery is historically considered the preferred management option for Type 3 g-NENs due to their higher rates of deep muscular invasion, lymphovascular involvement, and metastasis. However, with improved access to endoscopic and imaging diagnostic modalities, there is evidence of a shift in the stage of disease at the diagnosis of type 3 g-NENs. A recent comprehensive review of ten retrospective studies on type 3 g-NENs, including 229 patients, found G1 lesions in 66 and G2 lesions in 52 patients, suggesting that in the modern era, type 3 g-NENs are being diagnosed at an earlier stage [25]. This opens the opportunity to detect and definitively resect even type 3 g-NENs using minimally invasive endoscopic approaches, thus reducing the morbidity associated with surgery. Thus, ESD and potentially EMR could be the preferred approach to definitive management of well-selected patients with small (<20 mm), low-grade (G1/G2) type 3 g-NENs with no evidence of lymphovascular invasion or distant metastasis. While advanced endoscopic resection techniques have shown promising outcomes with very low recurrence rates (<5% with ESD), it is essential to note that simple polypectomy is not an ideal approach given reported recurrence rates higher than 50% [25].

### 2.3. Small Intestine

Based on a study using Surveillance, Epidemiology, and End Results (SEER) data from 1973 to 2012, the maximal increase in the incidence rate of NENs is for small intestinal NENs (0.8 per 100,000) with 38% of all gastrointestinal NENs found in the small intestine [3]. Small bowel NENs can be broadly conceptualized into two groups, those in the duodenum (d-NENs) and those in the rest of the small bowel, as these entities behave differently. One of the potential explanations for this is that duodenal tumors are easily identified using standard endoscopic techniques, while endoscopic access to the jejunum and ileum remains challenging. d-NENs also are identified incidentally on upper endoscopies that may be performed for other indications, which can result in earlier detection, while jejunum and ileum are not typically accessed on routine endoscopies. Duodenal NENs include gastrinomas, somatostatinomas, nonfunctional NENs, gangliocytic paragangliomas, and poorly differentiated neuroendocrine carcinomas [34]. Periampullary NENs and NENs of the ampulla of Vater are also duodenal, but they are often considered to be a separate class by many experts given their distinct histological, immunohistochemical, and growth behavior [34]. Most d-NENs are located in the first or second part of the duodenum, with about 20% occurring in the periampullary region [35]. Duodenal NENs can be asymptomatic or produce symptoms due to local infiltration, resulting in pain, gastrointestinal bleeding, and jaundice due to biliary obstruction or intestinal obstruction. Symptoms due to ectopic hormone release and classic carcinoid syndrome are rare (<10%) with duodenal compared to other small bowel NENs [34].

Neuroendocrine neoplasms (NENs) of the duodenum have also been categorized into ampullary and non-ampullary subtypes, depending on their anatomical location. Ampullary NENs, comprising 1/5th of duodenal NENs, exhibit distinct clinical characteristics when compared to non-ampullary NENs [36]. Ampullary is commonly observed in younger patients and frequently associated with Neurofibramatosis-1. Moreover, they are generally larger in size and tend to present with obstructive jaundice [37].

The ENETS 2016 consensus guidelines recommend surgical resection for ampullary and periampullary NETs [27]. However, for smaller ampullary NENs with a diameter less than 1 cm, endoscopic papillectomy has shown promise as a successful treatment option, particularly for low-grade tumors (G1), after ruling out muscular and bile duct invasion [38,39]. Notably, endoscopic papillectomy has also been documented for ampullary NENs larger than 10 mm [40,41]. Interestingly, recurrence after endoscopic papillectomy has not been observed so far, although further long-term studies are needed to validate this finding.

Despite encouraging results of endoscopic papillectomy, surveillance intervals after endoscopic treatment for ampullary NENs remain to be established. For ampullary NENs greater than 1 cm in size, surgical resection with lymphadenectomy is generally recommended due to their higher likelihood of lymph node involvement [42]. This approach aims to achieve complete tumor removal and prevent potential metastatic spread.

Similar to g-NENs, d-NENs also arise from the deep mucosal or submucosal layers and endoscopically appear as subepithelial lesions that are hemispherical or flatly elevated with reddish or yellowish hue on high-definition white light endoscopy (Figure 2 and Figure 3) [43].

They also tend to be more prominent or steep compared to gastric NENs and, with increasing size, tend to form a central depression, which can ulcerate with further tumor progression. EGD with direct tissue biopsy is the most common method to diagnose d-NENs conclusively. Given the subepithelial nature of these lesions, multiple deep bite-on-bite or tunnel biopsies are required to avoid missing the tumor tissue. EUS can accurately evaluate for the depth of tumor invasion and evidence of lymphovascular involvement. Unlike type-1 g-NENs < 10 mm, where surveillance can be an option, resection is recommended for all d-NENs independent of the size as there is no evidence that surveillance is a safe strategy as lymph node metastasis and microvascular invasion is observed even with small d-NENs [23]. ESGE recommends surveillance only for patients who refuse surgery or have severe comorbidities precluding them from undergoing resection. In such cases where surveillance is chosen, EUS is the recommended modality as it can best assess for various high-risk features [23]. EUS surveillance at 3–6 months then at 6–12-month intervals for 20–30 mm lesions, at 1–2 years for 10–20 mm lesions, or 2–3 years for lesions < 10 mm is a suggested strategy by ESGE [23]. The ideal resection strategy depends on factors like tumor size, location (periampullary vs. non-ampullary), and evidence or suspicion of metastatic disease. d-NENs that are small (≤10 cm), non-ampullary, and without evidence of lymph node or metastatic invasion can be resected endoscopically. Large ones (>2 cm) are usually treated with limited surgical resection, while the management of intermediate-sized ones (1–2 cm) can be controversial. NCCN guidelines recommend endoscopic resection for well-localized d-NENs whenever possible [44].

Resection options for d-NENs include minimally invasive endoscopic resection (EMR and ESD), operative local resection with primary repair of the duodenum, or a more radical pancreaticoduodenectomy [45]. EMR and ESD are the two endoscopic resection techniques employed for this purpose. In addition to the standard EMR technique, modifications, including cap-assisted EMR, underwater EMR, and ligation-assisted EMR, have been described for the resection of d-NENs. Although effective, EMR may not achieve complete pathological resection for d-NENs that infiltrate the deep submucosa, in which case ESD would be the preferred resection strategy. In a meta-analysis of 10 studies involving 224 d-NENs, it was found that among various EMR techniques, cap-assisted EMR resulted in the maximum rate of resections with free margins attained in 71% of cases. ESD resulted in maximal technical and clinical success with 100% en bloc resection with negative margins and the lowest recurrence rate; however, this was associated with a higher rate of adverse events, including bleeding and perforation [46]. Risk of delayed bleeding after endoscopic resection can be reduced by prophylactic clip closure, including over-the-scope-clip or newer methods like applying hemostatic agents like gels and powders [47,48,49]. Tran et al. found no difference in survival between EMR and SR among 104 patients with d-NENs 1–2 cm. Endoscopic full-thickness resection (EFTR) with laparoscopic assistance is another technique that has been described for managing d-NENs with excellent results, although data are limited at this time [50]. ESD should be the preferred resection technique, particularly when endoscopic resection is chosen for d-NENs larger than 1 cm. ESD, however, is a technically demanding procedure, and duodenal anatomy can make it challenging. Hence, appropriate patient selection, based on available procedural expertise, could be the key to successful outcomes.

Jejunal and Ileal (midgut NENs) behave differently than d-NENs, are often associated with carcinoid syndrome, and are difficult to diagnose given the anatomic location, making endoscopic access challenging. They are also more likely to be multifocal. Midgut can also be the site of the primary tumor in metastatic NENs of an unknown primary. Cross-sectional imaging, including somatostatin receptor scintigraphy, can often fail to detect small midgut NENs, in which case wireless video capsule endoscopy (VCE) can be considered [51,52]. Although valuable, VCE can be suboptimal in many cases and also be affected by various factors resulting in false positive results like intestinal peristalsis and extrinsic compression. VCE also lacks the ability for air or water insufflation and cannot obtain tissue for diagnosis. VCE was shown to have a sensitivity of 75% and a specificity of 38% for the localization of the primary tumor in NENs with an unknown primary compared to exploratory surgery [52]. Balloon enteroscopy, performed by an experienced endoscopist using single- or double-balloon techniques, can offer access to the entire small bowel. It can provide accurate localization and a better estimate of the number of lesions prior to surgical exploration. In a study involving 85 patients, 45 underwent double-balloon enteroscopy (DBE) as part of their pre-surgical evaluation, and 28 (62.20%) were found to have additional small bowel lesions on DBE. The major limitation of DBE is often the lack of availability of endoscopists with adequate training to perform the procedure accurately and efficiently [53]. However, with the availability of training, an advanced endoscopist can utilize this technique to effectively evaluate small bowel NENs that are not accessible to standard endoscopic techniques. Endoscopic resection is currently not a recommended treatment option for midgut NENs, as even small (<5 mm) lesions have also been shown to have the potential for deeper invasion and lymphatic spread [23].

### 2.4. Appendix

Most appendiceal NENs are detected incidentally on appendectomy specimens. Seen in 0.3–0.9% of specimens, these are commonly located at the tip of the appendix [54]. However, when they develop at the appendiceal base and cause obstruction, they present as appendicitis in around 10% of the cases [55]. Careful endoscopic assessment of the appendix during regular colonoscopy can identify these lesions, which may be subtle and often mimic an inverted appendix. NCCB, NANETs, and ENTES recommend right hemicolectomy (RHC) for appendiceal NET ≥ 2 cm. Synchronous CRC must be ruled out by a complete colonoscopy before RHC [56,57,58]. Appendectomy is recommended for NET < 1 cm. Definitive guidelines regarding post-resection surveillance of NETs ≤ 2 cm are currently lacking. In NETs ≥ 2 cm, NCCN 2013 guidelines recommend a follow-up 3–12 months post-resection, followed by every 6–12 months up to 10 years, with imaging or lab markers [59].

### 2.5. Colon

Colonic NENs account for roughly 7.5% of all NETs in US-based series [60,61,62]. Their increased detection rates have been attributed to rising rates of screening colonoscopies, with a ten-fold increase in incidence per data from the US SEER database [63]. The mean age of diagnosis is 55–65 years, with African American preponderance [64]. Colonic NENs are frequently aggressive, poorly differentiated, and higher grade than rectal NENs [64]. They are commonly detected incidentally on screening colonoscopy. The lack of early symptoms means that nearly a third of these tumors have already metastasized by the time they are diagnosed [63]. Symptomatic individuals may present with diarrhea, abdominal pain, GI bleeding, or weight loss [60]. Among GI NENs, those originating in the colon have the worst prognosis, with 5-year survival between 40–70% [60,62,65].

It is essential to perform a complete thorough colonoscopy evaluation once a single colorectal NEN is identified to evaluate for any synchronous lesions. Colonic NENs were most commonly located in the cecum, appeared as sessile/submucosal lesions, and were larger than 2 cm in size in a population-based case series from the Netherlands [66]. They appear darker than the surrounding mucosa, with a more prominent vascular pattern. Staging EUS can be used to evaluate the size of the tumor and depth of invasion to determine therapy options. Colonic NENs are treated akin to adenocarcinomas. Endoscopic mucosal resection or polypectomy can be used to resect lesions < 2 cm without lymph node metastasis. Chen et al. utilized ESD for the resection of colon carcinoids in 6 of their 239 patients with colorectal carcinoids < 2 cm. They reported that though feasible, ESD for colonic carcinoids has a higher non-R0 resection rate and higher complications [67]. ESD has been shown to have superior R0 resection rates compared to EMR while avoiding the risks of surgery [68]. Colonic ESD is technically more challenging than rectal ESD, given poor scope maneuverability attributed to physiologic flexion, colonic peristalsis, and respiratory movements. The pocket-creation method (PCM) for colonic ESD was shown to be faster, with better en bloc and R0 resection rates than conventional ESD [69]. Longer procedure time, especially with larger lesions, is the main drawback for ESD over EMR or polypectomy.

The recently described double-balloon endoluminal intervention platform (DEIP), which provides a more stabilized dissection plane without the need for creating a pocket, may be utilized [70]. Underwater ESD has been shown to be more effective in en-bloc resection of larger lesions [71]. The European Neuroendocrine Tumor Society (ENETS) recommends oncological resection for incomplete resection or poorly differentiated tumors [63]. Unfortunately, most colonic NENs are invasive through the muscularis propria and >2 cm, necessitating localized colectomy with oncological resection of lymphatic drainage, as per the National Comprehensive Cancer Network (NCCN) guidelines [2].

Follow-up depends on tumor size and grade. The North American Neuroendocrine Tumor Society (NANETS) consensus guidelines recommend annual surveillance with cross-sectional imaging for patients with stage II or III tumors (invading into or beyond muscularis propria or involving locoregional lymph nodes). They comment that long-term endoscopic or radiographic surveillance is not justified for stage I tumors (submucosal, ≤2 cm). However, given their high risk of relapse, hindgut NENs must be followed for at least seven years post-resection [72]. Future studies may look into long-term recurrence rates post-ESD of colonic NENs and fully explore novel options like DIEP.

### 2.6. Rectum

Rectal NENs (r-NENs) are considered a different clinical entity than colonic NENs. Like colonic NENs, they are being increasingly detected primarily due to the increasing number of colonoscopies for colon cancer screening. Most (90%) are asymptomatic, small (<20 mm), well differentiated, confined to the submucosal layer, and detected incidentally on endoscopic evaluation. Less commonly, they can present with symptoms such as rectal bleeding, change in bowel habits, or anal discomfort [73,74]. Endoscopically, they appear as small round polypoid subepithelial lesions characterized by smooth, normal-appearing, or yellowish mucosa. These typically tend to be grade 1 on the WHO classification. Larger and more advanced r-NENs can have varied appearances, including being semi-pedunculated with the absence of a pit pattern and areas of amorphous appearance with central erosions or ulcerations [74]. NBI can be a valuable tool to characterize these atypical lesions as described by Veyre et al. [74]. r-NENs are usually located about 4–10 cm above the dentate line in the mid-rectum. As with NENs in other locations of the GI tract, EUS is now recommended for the staging of most r-NENs, perhaps with the exception of very small (<5 mm) lesions [64].

Standard polypectomy results in unacceptable rates of incomplete resection and is not recommended for managing r-NENS [64]. Endoscopic treatment options include EMR (including modified EMR techniques like band-, cap-assisted EMR, and underwater EMR), ESD, transanal endoscopic microsurgery (TEMS), and EFTR [64,75]. Endoscopic resection is the preferred treatment option for all r-NENs < 1 cm with no evidence of muscularis propria invasion or local lymphovascular or metastatic spread. Conventional EMR is a very effective resection technique for r-NENs less than 1 cm, and studies have shown en bloc resection rates of up to 99%. However, histologically complete resection can be a challenge, with rates between 55 and 75% [76,77]. It is unclear whether histologically incomplete resection has clinical significance, and most studies report recurrence rates of 0 to 2% after EMR for these small lesions [77,78]. Modified EMR techniques have been developed over the past few years, which have substantially improved the rates of en bloc and R0 resection even with larger size r-NENs. As discussed earlier in this paper, these techniques include C-EMR, U-EMR, L-EMR, and hybrid-EMR. In a study evaluating L-EMR for small (<1 cm) d-NENs, Bang et al. reported an en bloc and R0 resection rates of 100% [79]. Similarly, from a study involving 114 patients, Park et al. reported a 100% endoscopically complete resection rate with 92.30% histological complete resection using C-EMR for r-NENs < 10 mm [80]. A meta-analysis involving 11 studies with 811 patients evaluating the efficacy and safety of modified EMR in r-NENs showed a significantly higher rate of histologically and endoscopically complete resection among patients treated with m-EMR than those treated with conventional EMR [81]. This study showed a higher vertical margin involvement rate with conventional EMR compared to m-EMR, while lateral margin involvement was not statistically significant.

ESD offers the advantage of higher en bloc resection rates compared to EMR, irrespective of lesion size. ESD was reported to be superior to C-EMR for larger (7–16 mm) r-NENs with higher rates of pathologically complete resection (100% vs. 70%) in a study comprising 55 patients [82]. Studies evaluating appropriate resection techniques for smaller (<10 mm) r-NENs found that there was no significant difference in outcome when m-EMR techniques are compared to ESDs. Huang et al. randomly assigned 50 patients with small r-NENs to double-band ligation-assisted EMR or ESD and found that the rate of en bloc and pathologically complete resection was 100% in both groups with no significant difference in rates of complications [83]. Other studies have also noted no significant difference in the outcomes of patients with small r-NENs treated with ESD compared to EMR with shorter procedure time in the EMR group [83,84]. A systematic review and meta-analysis of 14 studies showed that EMR with suction was superior to ESD in terms of higher complete resection rate, shorter procedure time, and similar overall complication rate and recurrence [85]. Thus, endoscopic resection is a very effective method to treat early small r-NENs (<10 mm), and the resection technique should be carefully selected based on indications and local expertise. Lesions 10–20 mm pose a greater challenge with a 5–15% risk of metastatic disease [86]. They can be considered a locally resectable advanced disease when there is no evidence of deeper muscular invasion or lymph node involvement of distant metastasis. r-NENs > 20 mm have up to 80% risk of distant metastasis and are treated surgically.

ESD and transanal endoscopic microsurgery (TEM) are the two minimally invasive techniques that can be used to resect these lesions. TEM is a minimally invasive surgery that allows full-thickness resection of rectal lesions [87]. It is performed under general anesthesia, involves a rigid rectoscope, and uses anal retractors to dilate the anal sphincter and maintain exposure. A multichannel transanal device capable of maintaining endoluminal pressure is used for this procedure [88]. TEM offers the advantage of deeper vertical resection margins, but single-center experiences have reported longer operative times than ESD, higher rates of postoperative morbidity following anal dilation, anesthesia-related adverse events, and more extended hospital stays. Jin et al., in their single-center retrospective study on 114 patients with r-NENs, reported that although the rate of complete resection in the TEM group was higher than ESD group, there was no difference in the rate of recurrence between the two groups in long-term follow-up [89]. Park et al. also reported a better overall R0 resection rate with TEM but no difference in recurrence rate compared to ESD, but the TEM group had a higher mean procedure time and hospital stay [90]. ESD could be the preferred resection technique for primary r-NENs, with TEM reserved for incompletely excised or recurrent lesions where fibrosis in the submucosal layer may limit ESD. Endoscopic full-thickness resection (EFTR) using a full-thickness resection device (FTRD) is a new promising endoscopic resection technique for managing r-NENs. Meier et al. reported macroscopically and histologically complete resection in all 40 cases (100%) of r-NENs treated using this technique [91]. In another study comparing EFTR to TEM for the treatment of r-NENs, the rates of en bloc resection rate, R0 resection rate, tumor size, and specimen size were similar, but there was a significant difference in the mean procedure time (48.9 min in TEM group vs. 19.2 min in the EFTR group) [75].

### 2.7. Pancreas

Pancreas has both exocrine and endocrine functions, with its endocrine tasks carried out by hormones made within the islet cells. These include insulin, glucagon, somatostatin, ghrelin, and pancreatic polypeptide. Tumors originating from these pancreatic islet cells are a heterogenous group, including pancreatic neuroendocrine tumors (pNETs) and neuroendocrine carcinomas [92]. Tumors that result in excess hormone production and resulting symptoms are called functional tumors. However, most (75–90%) of PNETs are nonfunctional. Historically functional PNETs have been noted to have better prognosis, likely explained by earlier detection as they tend to be symptomatic.

EUS is a powerful tool in the diagnosis, histological grading, and staging of PNETs. EUS-guided FNA and, preferably, FNB can be used to diagnose or confirm suspected PNETS seen on other imaging modalities. EUS-FNA has been the procedure of choice for obtaining cytological material for pathological analysis. EUS-FNB differs from EUS-FNA in its ability to obtain histological-quality core samples. This provides sufficient material to the pathologists to make a firm diagnosis and accurately grade and subtype PNETs using immunohistochemistry. FNB is more likely than FNA to provide adequate samples for Ki67/grading and showed a closer match to surgical histology [93]. EUS-FNB can hence reduce the rate of preoperative undergrading of PNETs. A recent meta-analysis found that the grading concordance between preoperative EUS samples and surgical specimens was significantly better with EUS-FNB compared to EUS-FNA (84.2% vs. 79.5%) [94]. Detecting somatic mutations using next-generation sequencing (NGS) is an emerging technique to identify tumors with a propensity for aggressive behavior and metastasis [95,96]. Mutations in MEN1 (menin), DAXX (death domain associated protein), ATRX (thalassemia/mental retardation syndrome x-linked), and mTOR (mammalian target of rapamycin) have been shown to occur with varying frequency in sporadic PNETs. The presence of these somatic mutations confers a higher risk for aggressive behavior [95]. Both EUS-FNA and EUS-FNB can provide tissue for NGS to identify these mutations. However, EUS-FNB again has been shown to have a significantly higher yield [95,96,97]. EUS can also assess for potential resectability by evaluating multifocality and lymphovascular involvement. A prospective study comparing the accuracy of preoperative diagnostic imaging using EUS, contrast-enhanced CT, and Gallium-DOTATOC positron emission tomography found that EUS had the highest specificity (98%) for evaluating lymph node metastasis [98]. Furthermore, recent advances in EUS techniques like contrast-enhanced EUS (CE-EUS) and contrast-enhanced harmonic EUS (CH-EUS) have highlighted characteristics with prognostic value that could aid complex clinical decision making in the management of PNETs [99]. The North American Neuroendocrine Tumor Society (NANETS) consensus paper on the surgical management of PNETs recommends EUS-FNB over FNA when available and recommends EUS imaging in diagnosing MEN 1 in addition to conventional imaging modalities. This paper also recommends EUS as an additional tool in informing surgical strategy [92].

Another emerging role of EUS in managing PNETs is EUS-guided ablation. EUS-guided radiofrequency ablation (EUS-RFA) and EUS-guided ethanol ablation (EUS-EA) are the two most well-described techniques [100,101]. While surgical resection is recommended for PNETS > 2 cm, there is considerable uncertainty in the management of very small (1 cm) and relatively small (1–2 cm) tumors. Based on evidence from retrospective studies, most of which are single-center case series, current guidelines suggest a wait-and-watch approach with consideration for resection based on the individual patient or tumor characteristics [92]. In a retrospective study from the Mayo Clinic that compared the clinicopathological features and outcomes of nonfunctioning PNETS < 4 cm who underwent surgical resection to those treated non-operatively, the authors observed no significant difference in disease-specific progression or mortality [102]. The median PNET size in this study was 1 cm in the non-operative group and 1.8 cm in the operative group. Another single-center retrospective case–control study comparing outcomes between surgically and non-surgically managed PNETS less than 3 cm also concluded that observation was a reasonable approach to small, stable asymptomatic PNETS [103]. However, other studies have shown evidence of aggressive behavior in small tumors.

Toste et al. reported a 7% rate of positive nodes for PNETS ≤ 2 cm that underwent resection, and Haynes et al. reported that 8% of small incidentally discovered nonfunctioning PNETS that were resected developed recurrence or metastasis [104,105]. A multicenter study from Europe involving 16 centers comprising 210 patients who underwent formal resections or enucleation for PNETs < 2 cm showed 10.6% positive nodes, again showing the potential for aggressive behavior in small PNETs [106]. This study also showed that the tumor grade influenced the node positivity rate, with 16% in grade 2 PNETs and 100% in grade 3 PNETs. In addition to the size, tumor histology and Ki-67 index have been recommended by the Canadian Expert National Group report while deciding on the appropriate management strategy for PNETS < 2 cm [107]. It is also essential to recognize that the current recommendations favoring a wait-and-watch approach for PNETS < 2 cm are based on the potential morbidity, mortality, and exocrine and endocrine deficiencies associated with surgical pancreatic resections. Endoscopic resection is associated with less morbidity and mortality, so the approach to these tumors could change in the future considering the overall risk-to-benefit ratio, particularly in the young and otherwise healthy patient population. EUS-guided ablation offers an attractive alternative option for managing these smaller PNETs, with studies showing excellent results while avoiding the potential risks of surgical pancreatic resection [108,109]. In a propensity score-matched retrospective study comparing the outcomes of EUS-guided ablation with surgical resection that included 285 patients, Ho et al. reported similar overall survival and disease-specific survival in both groups, while the EUS-guided ethanol ablation group had fewer adverse events and shorter hospital stays [110]. Another study comparing the outcomes of EUS-EA and surgery for small nonfunctioning pancreatic neuroendocrine tumors found that EUS-EA had fewer adverse events and shorter hospital stay with similar overall survival and disease-free survival rates compared to surgery [110]. In a systematic review and meta-analysis comparing EUS-RFA to EUS-EA in the management of PNETs, Garg et al. reported that the outcomes were similar. This included PNET sizes ranging from 10 mm to 27.5 mm and a combination of functioning and nonfunctioning PNETs.

### 2.8. Pulmonary

Pulmonary carcinoid tumors are low-grade malignant NETs [111]. They account for 1% of all primary lung cancers [112]. Peripheral carcinoids are generally <2 cm and seen as a single lobulated lesion on cross-sectional imaging [113]. CT-guided biopsy has poor yield and carries a risk of pneumothorax.

Pulmonary carcinoids typically metastasize to the hilar and mediastinal lymph nodes [114]. Mediastinal lymphadenopathy can sometimes be seen as extrinsic compression of the esophagus on EGD. The Commonwealth Neuroendocrine Tumor research collaborative (CommNETs) and NANETs recommends a tumor location-based biopsy approach [115]. While bronchoscopy can be utilized for central tumors, a transbronchial or transthoracic approach may be needed for peripheral tumors. In the case of metastatic carcinoid, the most accessible site, generally mediastinal lymphadenopathy, must be biopsied. In such cases, endosonography can be employed as a first-line approach, avoiding the risks and costs of surgical staging [116].

Accurate staging is essential for planning therapy for lung cancer. Histological confirmation of mediastinal adenopathy is needed to confirm metastatic disease. EUS with FNA/FNB and endobronchial ultrasound (EBUS) are minimally invasive, less expensive options with lower morbidity compared to surgical options like mediastinoscopy or thoracoscopy for the staging of mediastinal adenopathy [117]. EUS can help locate and direct FNA from adenopathy in the posterior and inferior mediastinum for paraesophageal, posterior, and inferior mediastinal lymph nodes. Endobronchial ultrasound (EBUS), on the other hand, is useful for anterior mediastinal adenopathy, including the paratracheal lymph nodes [118]. EUS-FNA has a sensitivity close to 90% when used in patients with suspected lung cancer and mediastinal nodes ≥ 1 cm in the short axis [119]. The diagnostic accuracy for mediastinal staging in lung cancer increases by using a combination of EUS-FNA and EBUS-trans bronchial needle aspiration (TBNA) and is recommended by the European Societies [120]. Hepatic metastases are common in lung carcinoids, and EUS can help identify suspected liver metastases and tissue acquisition, sometimes not seen on imaging [121,122].

## 3. Limitations

Advanced endoscopic procedures are associated with a learning curve. A recently published study reported that it takes around 100 colorectal procedures to achieve proficiency in advanced colorectal endoscopic procedures [123]. Given the need for dedicated equipment and longer procedure time, performing these may be feasible only in tertiary or quaternary centers. In the end, it comes down to the skill and experience of the advanced endoscopist. Advanced techniques carry a higher risk of complications such as bleeding and perforation compared to traditional endoscopic techniques. For instance, a meta-analysis analyzing complications of colorectal ESD reported a rate of 4.2% for immediate perforation, which is much higher than the conventional colonoscopy [124]. Longer procedure time puts the patient at risk for complications related to anesthesia.

Given the rarity of GI NETs, they lack clear guidelines for advanced endoscopic management. As advanced endoscopic techniques continue to evolve, their safety and efficacy in managing NETs must ideally be validated on large patient cohorts, though this may not be possible given the uncommon occurrence of these lesions. A good starting point for further research may be elucidating precise limitations for advanced endoscopic procedures along the length of the GI tract so that a prompt surgical referral can be made. Risk-stratifying tools that account for lesion morphology on cross-sectional and endoscopic imaging can be developed to steer the management of these NETs in the correct direction—whether it is towards or away from the advanced endoscopist.

## 4. Conclusions

NENs, which were once a very rare condition, are now being diagnosed with increasing frequency and at an earlier stage in their natural course. Although traditionally considered indolent in a majority of cases, it carries the risk of malignant transformation and distant metastasis. Accurate diagnosis and early appropriate management while limiting the morbidity and potential mortality associated with interventions are key to a favorable outcome. Advanced endoscopic techniques are minimally invasive, safe, and effective options for the comprehensive evaluation and management of NENs.

## Figures and Tables

**Figure 1 cancers-15-04175-f001:**
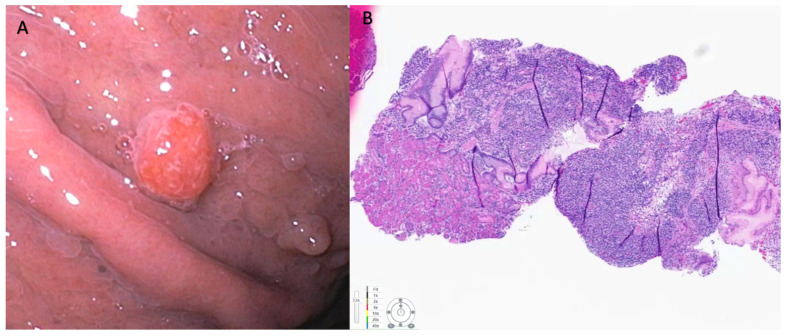
(**A**) Neuroendocrine tumor with smooth reddish overlying mucosa noted in the gastric antrum measuring 0.7 cm. This was resected using EMR technique. (**B**) Pathology showed negative margins. H and E staining. Immunostaining for Ki-67 showed 5% staining that is consistent with a diagnosis of grade 2 neuroendocrine tumor.

**Figure 2 cancers-15-04175-f002:**
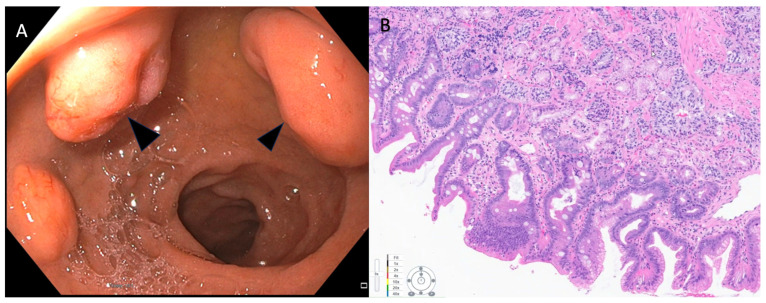
(**A**) Endoscopic appearance of sessile polyps noted in the duodenal bulb that were suspicious for duodenal NET (Arrow heads). They were both about 2.2 cm in maximal diameter. (**B**) Pathology from endoscopic bite-on-bite biopsies was consistent with well-differentiated neuroendocrine tumor. Patient underwent surgical resection using robotic sleeve duodenectomy. Pathology of surgical specimen confirmed grade-1, well-differentiated neuroendocrine tumor.

**Figure 3 cancers-15-04175-f003:**
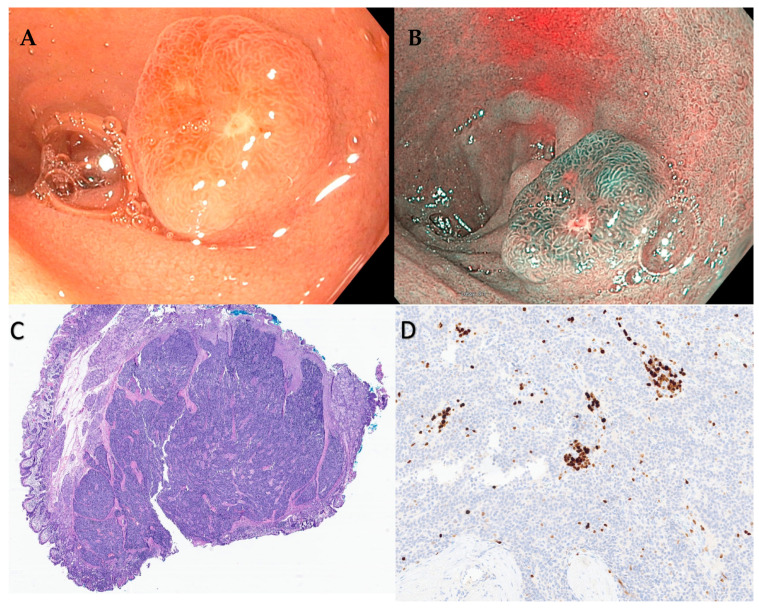
(**A**) Duodenal neuroendocrine tumor measuring about 1.2 cm. (**B**) NBI view of the same lesion. This was resected using EMR technique. (**C**) H and E staining showing NET. (**D**) Immunostaining for Ki-67 showed <3% staining that is consistent with a diagnosis of grade 1 neuroendocrine tumor.

**Table 1 cancers-15-04175-t001:** WHO 2019 classification and grading criteria for Neuroendocrine neoplasms (NENs) of the GI tract and hepatopancreatic biliary organs; NET, Neuroendocrine tumor; NEC, Neuroendocrine carcinoma; SCNEC, Small-cell neuroendocrine carcinoma; LCNEC, Large-cell neuroendocrine carcinoma; MiNEN, Mixed neuroendocrine-non-neuroendocrine neoplasm.

Terminology	Differentiation	Grade	Mitotic Rate	Ki-67 Index
NET, GI	Well differentiated	Low	<2	<3%
NET, G2		Intermediate	2–20	3–20%
NET, G3		High	>20	>20%
NEC, small-cell type (SCNEC)	Poorly differentiated	High	>20	>20%
NEC, large-cell type (LCNEC)			>20	>20%
MiNEN	Well or poorly differentiated	Variable	Variable	Variable

**Table 2 cancers-15-04175-t002:** Possible endoscopic interventions based on location of neuroendocrine neoplasm. Note: CE-EUS, Contrast-enhanced Endoscopic ultrasound; FNA, Fine-needle aspiration; FNB, Fine-needle biopsy; SINK, Single-incision with needle knife; MAIB, Mucosal incision-assisted biopsy; ER, Endoscopic resection; ESD, Endoscopic sub-mucosal dissection; EMR, Endoscopic mucosal resection; STER, Submucosal tunnelling endoscopic resection; EFTR, Endoscopic full-thickness resection; VCE, Video capsule endoscopy; PCM, Pocket-creation method; DEIP, double balloon endoluminal intervention platform; NBI, narrow-band imaging; TEMS, trans-anal endoscopic microsurgery; FNB, Fine-needle biopsy; RFA, Radio-frequency ablation; EA, Ethanol ablation; EBUS, Endobronchial ultrasound; TBNA, Trans-bronchial needle aspiration.

Location of NEN	Recommendation for Observation	Endoscopic Diagnostic and Therapeutic Techniques			Endoscopic Management Comments
		Diagnostic and Staging Techniques	Therapeutic Interventions First Line	Therapeutic Interventions Second Line	
Esophagus	Currently there are no treatment guidelines that specifically address esophageal NENs. Treatment choice should be based on assessment of tumor size, grade, stage, patient’s coexisting health conditions and local expertise	EGD ± NBI with biopsy EUS and CE-EUS EUS-FNA, FNB SINK MIAB	ESD	EMR Enucleation with SMT	Endoscopic resection recommended only for lesions < 1 cm with no suspicion for lymphovascular invasion
Stomach	Surveillance could be an option for Type-1 g-NENs < 1 cm. Treatment choice should be based on assessment of patient’s coexisting health conditions and local expertise. Observation is not favoured for Type-2 and Type-3 n-NENs. Local or limited excision can be considered, but must be tailored to the patient based on multidisciplinary evaluation at centers with expertise.	EGD ± NBI with biopsy, E-EUS EUS-FNA, FNB SINKMIAB	ESD	EMR EFTR Underwater EMR	Endoscopic resection recommended only for lesions < 2 cm with no suspicion for lymphovascular invasion
Small Intestine	Observation is not recommended for d-NENs or for NENs originating in the jejunum or ileum as they have a higher potential for an aggressive behaviour. Furthermore, endoscopic access to distal small bowel is limited and precludes effective surveillance.	EGD ± NBI with biopsy EUS EUS-FNA, FNB VCE Single or Double balloon enteroscopy	EMR (cap-assisted, underwater, ligation-assisted) ESDEFTR	No consensus. Based on local expertise, location and features of NEN	Endoscopic resection recommended only for lesions < 1 cm with no suspicion for lymphovascular invasion. NENs between 1 and 2 cm can be considered for endoscopic resection on a case-by-case basis.
Colon	Observation is not recommended. Small colonic NENs are often mistaken for hyperplastic polyps on colonoscopy and get resected using cold snare polypectomy.	Colonoscopy ± NBI with biopsy	EMR ESD ESD using DEIP		Endoscopic resection recommended only for lesions < 2 cm with no suspicion for lymphovascular invasion
Rectum	Observation is not recommended as these are easily accessible to endoscopic resection.	Colonoscopy ± NBI with biopsy EUS EUS-FNA, FNB	EMR (ligation-assisted, hybrid, modified) ESD	EMR (cap-assisted, underwater, hybrid, modified) ESD EFTR	EMR is the preferred technique for removing rectal NENs < 10 mm in size, and ESD for lesions up to 20 mm. Transanal endoscopic microsurgery can also be an option for lesions> 10 mm but <20 mm.
Pancreas	Surveillance can be an acceptable strategy for asymptomatic patients with a pNETs < 1 cm. Decision to observe or resect an asymptomatic pNET 1–2 cm in size should be individualized based on patient age, co-morbidities and tumor behavior during surveillance. Well informed decision from the patient and access to dependable long-term follow-up will also be important factors to consider.	EUS-FNA, FNB	EUS-RFA EUS-EA		Endoscopic ablation techniques could be an effective treatment option for patients who would otherwise be considered for observation. One of the most important factor in the decision for observation instead of surgical resection is the relatively high morbidity and mortality associated with pancreatic surgeries.

**Table 3 cancers-15-04175-t003:** Summary of North American Neuroendocrine Tumor Society (NANETS) and the European Neuroendocrine Tumor Society (ENETS) recommendations on endoscopic management of NENs. NEN, Neuroendocrine neoplasm; NANETS, North American Neuroendocrine Tumor Society; ENETS, European Neuroendocrine Tumor Society; EUS, Endoscopic ultrasound; FNA, Fine-needle aspiration; FNB, Fine-needle biopsy; ER, Endoscopic resection; SR, Surgical resection; FTR, Full thickness resection; LAR, Low anterior resection; APR, Abdominoperineal resection; RFA, Radiofrequency ablation; PNET, Pancreatic neuroendocrine tumor.

Location of NEN	NANETS Guidelines	ENETS Guidelines
Esophagus	No consensus.	No consensus.
Stomach	Type 1 <1 cm Surveillance or ER 1–2 cm Endoscopic surveillance every 3 years vs. ER >2 cm SR. ER can be considered if feasible on a case-by-case manner EUS recommended for NEN > 1 cm to assess depth of invasion Type 2 <1 cm Surveillance or ER 1–2 cm ER or SR >2 cm SR EUS recommended for NEN > 1 cm to assess depth of invasion Type 3 Surgical resection	EUS is recommended in tumors >1 cm Endoscopic resection in type 1 NEN larger than 1 cm. ESD and FTR are more effective to achieve R0 resection compared to EMR. Endoscopic resection may be considered for localized type III G1 gNETs ≤ 10 mm, and occasionally larger tumors with Ki-67 < 10% and <15 mm in diameter if the risks of surgical resection are high.
Small Intestine	No definite recommendations for ER in duodenal NENs, but guidelines state that ER is potentially appropriate for localized tumors <2 cm, if surgically feasible. Surgical resection with lymph node dissection and surgical full bowel examination to evaluate for lateral metastasis in NENs of jejunum, ileum and cecum. Appendiceal NEN <1 cm SR 1–2 cm SR/ Right hemicolectomy. >2 cm Right hemicolectomy	Biopsies to confirm the diagnosis and for grading EUS is recommended in tumours >1 cm Very small non-functioning tumors in D1 should be removed using ER Lesions of 5–10 mm (and up to 15 mm in some centers) can be removed endoscopically after imaging work-up, but risks are relatively high
Colorectum	<1 cm, ER 1–2 cm, ER or SR >2 cm, ER or SR	≤10, ER is recommended and recurrence rates are low. ≥20 mm, SR using LAR or APR is recommended (after exclusion of unresectable distant metastases). 10–20 mm, MDT discussion about either endoscopic or surgical therapy.
Pancreas	EUS-FNA should be performed for diagnosis or when there is a question about tumor grade. Although FNA is most frequently performed, the addition of FNB can be performed where available. EUS should be performed to identify multifocal disease in MEN1 patients. EUS does not need to be performed to determine surgical resectability.	EUS for evaluation of all PNETS with EUS guided tissue sampling whenever possible. Guidelines mention that EUS directed ablation using ethanol injection or CT-guided RFA have all been successful for PNETS < 2 cm (insulinomas and MEN1), but no specific recommendations.

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
