# Peer review of "Role of Advanced Gastrointestinal Endoscopy in the Comprehensive Management of Neuroendocrine Neoplasms"

_cancers, 2023, doi:10.3390/cancers15164175_

Round 1

Reviewer 1 Report

well done and interesting paper to read and to know about 

Author Response

We greatly appreciate your time and feedback. Thank you very much for reviewing our manuscript. 

Reviewer 2 Report

Th authors have performed an exhaustive review of the literature about the NEN of the GI tract and pulmonary tract.

They have shown the last techniques in the diagnosis and therapy.

Author Response

(The authors gave the same response as above.)

Reviewer 3 Report

This is a nice overview of the role of endoscopy in the management of neuroendocrine neoplasms. Here are my comments to improve the manuscript:

1) Please, try to provide a table resuming the recommendations for the management (observation, resection, etc) of GI NEN according to the location (stomach, duodenum, etc)

2) Please provide more details on the management of ampullary NEN by creating a specific paragraph. 

3) Please, implement the paragraph on pancreatic NEN. In particular, the performance of EUS-FNA and EUS-FNB should be more in deep discussed. Similarly, the possibility to obtain new prognostic markers (DAXX/ATRX) on EUS-FNB specimens, the performance of EUS in detecting positive lymphnodes, and the performance of EUS-guided ablation in comparison with surgical resection should be mentioned. Here I suggest some papers that could be included: PMID 37169669, PMID 35986682, PMID 31579710, PMID 36871765, PMID 37488390, PMID 32783272, PMID 34819995, PMID 35863518, PMID 35930017, PMID 36400239

4) It would be nice to have one table reporting the NANETS and ENETS recommendations.

Author Response

We greatly appreciate your time and feedback. Thank you very much for reviewing our manuscript. Please find below your comments/suggestions and our point-by-point response to those. 

Reviewer
1) Please, try to provide a table resuming the recommendations for the management (observation, resection, etc) of GI NEN according to the location (stomach, duodenum, etc)

Answer: Thank you very much for this excellent suggestion. We have added additional columns to Table 2 with recommendation regarding endoscopic resection and observation per your recommendations. As discussed in this manuscript the overarching reason behind observation for small (< 1 cm) NETs is the balance between potential adverse events of surgical intervention Vs. the relatively low risk of progression or aggressive behavior seen with these tumors. However, the risk of progression is not negligible. With the advances in endoscopic resection techniques and the very low risk of adverse events, we focus on the potential for minimally invasive endoscopic resection or ablation even for smaller lesions whenever the expertise is available.

2) Please provide more details on the management of ampullary NEN by creating a specific paragraph. 

Answer: Thank you very much for your input. We have added this section under the discussion on duodenal NENs.

3) Please, implement the paragraph on pancreatic NEN. In particular, the performance of EUS-FNA and EUS-FNB should be more in deep discussed. Similarly, the possibility to obtain new prognostic markers (DAXX/ATRX) on EUS-FNB specimens, the performance of EUS in detecting positive lymph nodes, and the performance of EUS-guided ablation in comparison with surgical resection should be mentioned. Here I suggest some papers that could be included: PMID 37169669, PMID 35986682, PMID 31579710, PMID 36871765, PMID 37488390, PMID 32783272, PMID 34819995, PMID 35863518, PMID 35930017, PMID 36400239

Answer: Thank you very much for your recommendation and for suggesting excellent references. This section has been updated per your recommendations.

4) It would be nice to have one table reporting the NANETS and ENETS recommendations.

Answer: Thank you very much once again for your suggestion. Please see the new table (Table 3) added to the revised manuscript,

Round 2

Reviewer 3 Report

I have no further comments